# Characterization on the Copolymerization Resin between Bayberry (*Myrica rubra*) Tannin and Pre-Polymers of Conventional Urea–Formaldehyde Resin

**Jinda Peng** [1], **Xinyi Chen** [1], **Jun Zhang** [2], **Hisham Essawy** [3], **Guanben Du** [1,2,*] **and Xiaojian Zhou** [1,2,*]

1   Yunnan Provincial Key Laboratory of Wood Adhesives and Glued Products, Southwest Forestry University, Kunming 650224, China; pjd1212@126.com (J.P.); xinyi.chen@univ-lorraine.fr (X.C.)
2   Key Laboratory for Forest Resources Conservation and Utilisation in the Southwest Mountains of China, Southwest Forestry University, Ministry of Education, Kunming 650224, China; zj8101274@163.com
3   National Research Centre, Department of Polymers and Pigments, Cairo 12622, Egypt; hishamessawy@yahoo.com
*   Correspondence: gongben9@hotmail.com (G.D.); xiaojianzhou@hotmail.com (X.Z.)

**Abstract:** By focusing on the disadvantages of weak water resistance and high formaldehyde emission of urea–formaldehyde resin (UF), this research provides a new method to overcome these shortages of UF resin by using tannin for partial substitution of urea. Furthermore, plasma pretreatment of wood was introduced to strengthen the bonding performance of plywood. The investigation of the chemical structure of UF resin and tannin–urea–formaldehyde resin (TUF) were performed with Fourier transform infrared spectroscopy (FT-IR) and solid-state $^{13}$C nuclear magnetic resonance ($^{13}$C NMR). The results of investigations confirmed the joining of tannin into the resin structure, which may enhance structural rigidity of TUF adhesives and improve hydrolysis stability. Then, thermal performance of UF resin and TUF resins were tested by differential scanning calorimetry (DSC) and thermogravimetric (TG) analysis. The DSC results indicated that the curing temperature did not change significantly. However, the TG analysis showed that the thermal stability of TUF resin was considerably improved. In bonding performance test, tannin–urea–formaldehyde resin (TUF) revealed an excellent water resistance, comparable to UF resin and can fulfill the standard requirement for plywood (Type II according to the Norm GB/T 17657-2013). It is interesting that the shear strength of wood specimens, bonded with TUF6 resin, after low-pressure cold plasma equipment (CLP plasma) and jet type atmospheric low-temperature plasma (JTLP plasma) treatment, reached 0.80 MPa and 0.85 MPa, respectively, after being soaked in boiling water for 3 h. In addition, most of the bonded plywood samples with TUF resin exhibited a lower formaldehyde emission, especially those prepared at 70 °C and 1.5 h, in which the formaldehyde emission amount could be reduced by approximately 39%.

**Keywords:** UF resin; bayberry tannin; formaldehyde emission; plasma techniques; water resistance

## 1. Introduction

The wood adhesive market research reported that urea–formaldehyde resins (UF) production reached around 12.3 million tons in 2020 all over the world, and it had been projected that the production of UF resin will grow at an annual rate of over 4% between 2021 and 2026 [1]. Urea–formaldehyde resin is a type of amino resin that can be manufactured by condensation reaction of formaldehyde with urea [2]. It has been especially used as a binder for interior wood-based composites, such as particleboard, plywood, medium-density fiberboard, etc., due to its low cost, low cure temperature, outstanding strength, low-toxicity, and other favorable properties [3,4]. Nevertheless, UF resin still presents some significant issues, for instance, formaldehyde emission, ease of aging, and poor water resistance; as such, its broadly exterior applications are limited as compared

with typical phenolic resins [5]. Some chemical modification approaches have been used to enhance the performance of conventional UF resins to suit applications in exterior environment [6–8]. The typical pathways are co-polycondensation to introduce melamine and phenol for melamine–urea–formaldehyde [9–11] and phenol–urea–formaldehyde co-polycondensation resin adhesives [12–15], respectively. Nevertheless, the high price or toxicity of raw materials will inevitably increase the cost or environmental risk either during the synthesis or practical applications [16]. Therefore, the researchers have given much attention to resolving the problem of high price or toxicity of starting materials by substitution with materials from natural resources.

Typical biomass resources, such as lignin, cellulose, tannin, etc., have been utilized for a long time to prepare or replace petroleum-based raw materials needed for production of wood adhesives [17–19]. As a natural polyphenol component, tannin, which comes next to lignin based on its reserve, can be collected from plants [20,21]. It has been extensively utilized to replace petroleum-based phenol totally or partially in typical phenol–formaldehyde resin, because of its similar chemical structure to phenol [22]. However, the properties of the modified resins have been weakened even though the environmentally friendly nature has been increased. Nevertheless, the tannin-modified phenol–formaldehyde resins hadbeen used to prepare the wood-based panels, such as plywood or particleboards, and their properties could be complied with requirements of exterior applications [23,24]. In addition to the replacement of phenol in phenol–formaldehyde resins, another viable application of tannin, which has gained a strong attention recently, is the development of non-isocyanate polyurethane resins for wood adhesives, based on tannin as a substitution of toxic isocyanates [25–28]. What is more, the formaldehyde can be removed completely in some formulations with the use of hexamine or some bio-based curing agents to incorporate the tannin into an environmentally friendly product [29,30].

Tannin was also used to modify typical urea–formaldehyde (UF) resins, due to the high reactivity of tannin and formaldehyde and its phenol-like structure, resulting in products such as plywood, particleboard, and fiberboard with low formaldehyde emission [31–35]. Generally, tannin is considered as an additional component that can be mixed with typical UF resins in different ratios [32–35]. Thus, products bonded with tannin–UF resins via blending modification can show a lower formaldehyde emission than those based on bindings undertaken by traditional UF resin. Nevertheless, the bonding performance of such products exhibited a drop with the tannin addition, whenthe tannin was not effectively linked within the chemical skeleton of the resin structure. Therefore, tannin was utilized with a UF resin system by introducing it during the preparation process to co-polymerize with formaldehyde and urea [31]. Apparently, this approach can enhance the bonding performance, in some extent, along with a decreased formaldehyde emission.

However, due to the difference of reactivity between A and B rings of tannin, together with its steric hindrance, it is hard to produce an effective cross-linked structure, but it exists as a short-chain or straight-chain structure. Thus, the molecular structure with a long chain is commonly used to overcome the steric hindrance between tannins, so that an effective connection between the tannins can be generated [36]. Herein, a novel tannin-modified UF resin was reported by substituting a part of the urea with tannin for preparing the tannin–urea–formaldehyde resin (TUF), i.e., tannin was directly added as an acidic agent to replace urea, which can cause participation in the resin synthesis during the last alkaline stage of the UF resin synthesis. The impact of the addition of tannin on reaching a state of effective co-polymerization with urea–formaldehyde prepolymer was investigated, which urea–formaldehyde prepolymer consists of monomethylolurea, dimethylolurea (most), and trimethylolurea that might be acting as a bridging agent between the tannin units, and avoidance of the classical disadvantages of UF resins, such as poor water resistance and high formaldehyde emission. In addition, different plasma surface modification regimes were adopted to pre-treat the wood veneer, which is expected to enhance the performance of plywood samples bonded with TUF resin.

## 2. Materials and Methods

### 2.1. Materials

Formaldehyde (37%, AR), urea (99%, AR), p-Toluenesulfonic acid (98%, CP), sodium hydroxide (AR), formic acid (88%, AR), and ammonium chloride (AR) were purchased from Sinopharm Chemical Reagent Co., Ltd. (Beijing, China). Additionally, where the AR and CP represented the analytical reagent and chemically pure, respectively. Bayberry tannin (72%) was purchased from Guangxi Wuming Quebracho plant, whose main structure is procyanidins and prodelphinidins. Commercial poplar veneers were bought in Nanjing with dimensions of 400 mm × 420 mm × 2 mm and moisture content between 8 to 10%.

### 2.2. Synthesis of UF Resin

The urea–formaldehyde resin was obtained based on the typical alkali–acid–alkali process [37]. Briefly, 100 g of formaldehyde was charged into a three-neck flask and the pH was adjusted at 8–9 using 20 wt% sodium hydroxide solution while the temperature was increased to 90 °C. Then, the first batch of urea (37 g) (U1) was added, and the pH was maintained around 8 to 9 for 30 min. After that, the pH of the above solution was adjusted to 4.5–5 by using 20 wt% formic acid solution to promote the polycondensation reaction until a cloudy state was attained. Finally, the pH of the mixture was adjusted to 8–9 again with 20 wt% sodium hydroxide solution; then, the second batch of urea (20 g) (U2) was added into the three-neck flask for additional reaction for 20 min, in which a final F/U molar ratio of 1.3 was obtained. Then, the mixture was cooled down to room temperature and stored for subsequent use.

### 2.3. Preparation of Bayberry Tannin-Modified UF Resin (TUF)

A series of TUF resins were obtained according to the formulations presented in Tables 1–3, which show the preparation of the resin under varied conditions.

**Table 1.** Reaction temperature as a variable for the synthesis of UF/TUF resins.

| Formulation | Reaction Temperature (°C) | Reaction Time (h) | U2 Addition (g) | Tannin Addition (g) |
|---|---|---|---|---|
| UF | 90 | 2 | 20 | 0 |
| TUF1 | 60 | 1 | 0 | 20 |
| TUF2 | 70 | 1 | 0 | 20 |
| TUF3 | 80 | 1 | 0 | 20 |
| TUF4 | 90 | 1 | 0 | 20 |

**Table 2.** Reaction time as a variable for the synthesis of TUF resins.

| Formulation | Reaction Temperature (°C) | Reaction Time (h) | U2 Addition (g) | Tannin Addition (g) |
|---|---|---|---|---|
| TUF5 | 70 | 0.5 | 0 | 20 |
| TUF2 | 70 | 1 | 0 | 20 |
| TUF6 | 70 | 1.5 | 0 | 20 |
| TUF7 | 70 | 2 | 0 | 20 |

**Table 3.** Tannin content as a variable for the synthesis of TUF resins.

| Formulation | Reaction Temperature (°C) | Reaction Time (h) | U2 Addition (g) | Tannin Addition (g) |
|---|---|---|---|---|
| TUF8 | 70 | 1.5 | 0 | 15 |
| TUF6 | 70 | 1.5 | 0 | 20 |
| TUF9 | 70 | 1.5 | 0 | 25 |
| TUF10 | 70 | 1.5 | 0 | 30 |

The first step of synthesizing TUF resin is the same as for UF resin. After the first alkaline stage, the temperature was rapidly reduced to 70 °C and the prepared tannin powder was added slowly into the solution without adjusting the pH. The condensation reaction was continued to definite times. Finally, the resulting resin was cooled to room temperature, and the final pH of the resulting resin was adjusted to 8–9 to be ready for use. The process of the synthesis of UF and TUF resins is demonstrated in Figure 1.

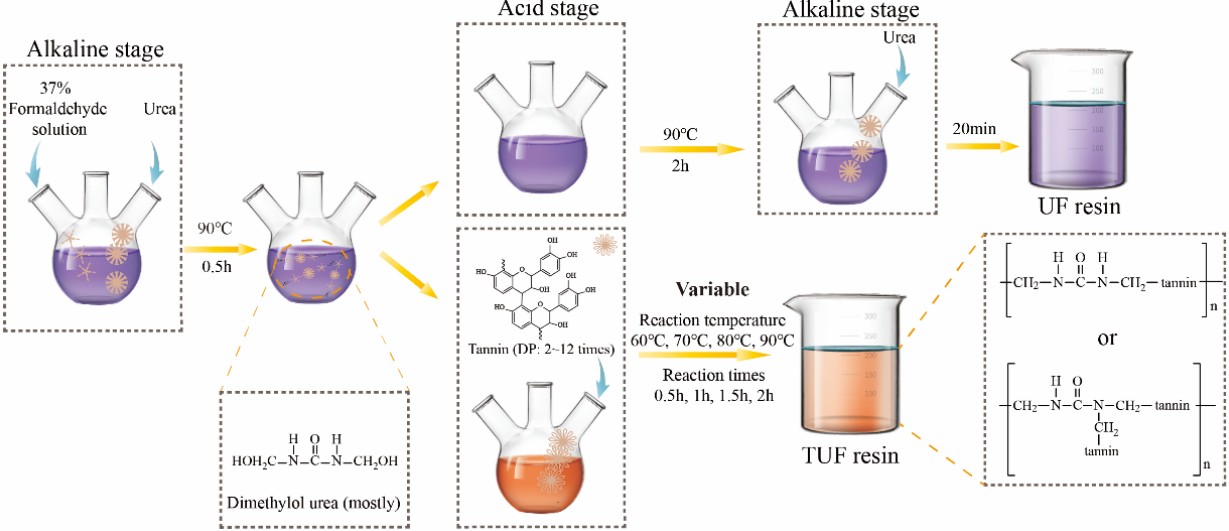

**Figure 1.** Synthetic steps for preparation of UF/ TUF resins.

In order to explore the influence of synthesis conditions on the performance of TUF resin, three-parameter experiments are set, whose variables are reaction temperature, reaction time, and tannin content as illustrated in Tables 1–3.

### 2.4. Evaluation of the Resin Properties

The fundamental performance of resins, including non-volatile content, viscosity, pH value, and gel time were specified according to the China national standard GB/T 14074-2017 [38] for testing.

The non-volatile content of the resins was measured by comparison of resin weight before and after drying. Firstly, about 1 g of resin was put into the oven at $120 \pm 1$ °C for 2 h, then the weight of the residue was recorded after cooling down to room temperature. The value of the non-volatile content was calculated as follows:

$$c = m/m_0, \qquad (1)$$

where, c is the non-volatile content of the resin, and m and $m_0$ are the weights before and after drying, respectively.

The viscosity measurements were conducted at 24 °C using rotational viscometer (SNB-2, digital viscometer, Shanghai Hengping Instrument Factory, Shanghai, China). The rotor and its speed were selected according to the viscosity of a resin sample. The pH values of the resins were determined using pH meter (FiveEasy Plus, METTLER TOLEDO, Shanghai, China). The gel time was measured at 100 °C by exposing 5 g of each resin in a test tube, into a boiling water and a wire was utilized to stir the resin constantly until hardening occurred. The tests were repeated three times for average calculation.

### 2.5. Investigations Using Fourier Transform Infrared (FT-IR) Spectroscopy

The resin structure was examined using infrared spectrometer (Thermo Scientific Nicolet iS50, Waltham, Massachusetts, USA). A drop of a liquid and untreated resin was placed on the surface of diamond crystal ATR accessory and scanned for 32 scans at 4 cm$^{-1}$ resolution over the range 500–4000 cm$^{-1}$.

### 2.6. Investigations Using Solid-State $^{13}C$ Nuclear Magnetic Resonance ($^{13}C$ NMR)

Solid-state $^{13}C$ NMR spectrometer, Bruker 400M (Zurich, Switzerland) was utilized for analysis of TUF resin with 4 mm dual-broad band CP-MAS probe at a frequency of 100.66 MHz. The spectrum was acquired by TOSS (total side band suppression) at ambient temperature with a spinning rate of 5 kHz.

### 2.7. Investigations Using Differential Scanning Calorimetry (DSC)

The curing behavior of the resins was investigated using DSC using NETZSCH DSC 200 F3, which the resin samples were freeze-dried after mixed with 6 wt% of p-toluenesulfonic acid (based on the dry weight), with the scans recorded at a heating rate of 15 °C/min under nitrogen atmosphere at a flow rate of 60 mL/min.

### 2.8. Investigations Using Thermogravimetric Analysis (TGA)

The thermal stability of the resins was tested using TGA analyzer model NETZSCH STA2500. The resin samples were put into an oven at 120 ± 3 °C until the samples were dehydrated and cured. The cured resin samples were ground to 200-mesh powder and approximately 5–8 mg was put in a platinum cup and heated under nitrogen atmosphere over the range from 40 to 600 °C at a heating rate of 10 °C/min.

### 2.9. Different Plasma Treatments for Activating the Surface of Veneers

The best experimental group was selected for veneer activation via plasma treatment according to the following regimes:

- Sliding cold arc plasma equipment (SAC plasma) was purchased from Nanjing Suman Plasma Technology Co., Ltd. in Nanjing, China and the parameters as follow [39]: power is 1000 W, the distance between the nozzle and the veneer was set at 3 cm, with a speed of treatment of 1 cm/s for a total treatment time of 1 min;
- Low-pressure cold plasma equipment (CLP plasma) was purchased from Changzhou Zhongke Normal Plasma Co., Ltd. In Changzhou, China and the parameters as follow [40]: power is 50 W, treatment was achieved under nitrogen atmosphere for 3 min;
- Jet type atmospheric low-temperature plasma surface treatment equipment (JTLP plasma) was purchased from Nanjing Suman Plasma Technology Co., Ltd. In Nanjing, China and the parameters as follow [41]: power is 500 W, single-board treatments back and forth for 3 min per side.

### 2.10. Preparation and Evaluation of Plywood

Three-layer plywood was prepared in the experiment by evenly coating the resin on both sides of the poplar veneer. The resin content was set at 280 g/m$^2$ on both sides. These veneers were stacked to form a veneer matrix based on the vertical texture of two adjacent veneers. Then, 2 wt% of 20% ammonium chloride solution, which based on the non-volatile content of the resin, was used as a hardener to make the glues. The hot press was accomplished at 120 °C for 5 min at a compression rate of 18%. Unlike the UF resin, the glues of TUF resins were formulated with 6 wt% of 40% p-Toluenesulfonic acid solution as a hardener, and the hot press was undertaken at 180 °C for 8 min at a compression rate of 18%. The shear strength test was carried out on plywood according to China National Standard GB/T 17657-2013 [42]. Different sets of the samples were soaked in cold water for 24 h, in a water bath set at 63 °C for 3 h and in boiling water for 3 h, separately. Seven specimens were tested in parallel for each case, and the average value and standard deviation were calculated.

### 2.11. Evaluation of Formaldehyde Emission

Formaldehyde emission from the prepared plywood panels was evaluated using a desiccator method based on the procedure described in China National Standard GB/T 17657-2013 [42]. A total of 24 samples (150 mm × 50 mm) were placed into a desiccator

under conditions of $20 \pm 2\ °C$ and relative humidity of $65 \pm 5\%$ for seven days. Then, the specimens were inserted into the specimen holder, and put directly above a glass dish filled with 300 mL of distilled water in desiccator to absorb emitted formaldehyde. After 24 h at $20 \pm 0.5\ °C$, the formaldehyde concentration in the solution was estimated spectrophotometrically using acetylacetone method.

## 3. Results

### 3.1. Basic Properties of Resins

The basic properties of TUF resin compared to UF resin are summarized in Table 4. In the table, due to the high reaction temperature and high tannin addition, the TUF4 and TUF10 were subject to gelation during the preparation process making it difficult to identify the resins characteristics, which is why TUF4 and TUF10 have no data in the table below. The non-volatile content of TUF resins did not change appreciably compared with UF resin and remained in the range of 53%~56%. This showed that tannin addition can maintain a suitable non-volatile content for TUF resins even though a partial substitution of urea was undertaken.

**Table 4.** Basic properties of TUF resins compared with UF resin.

| Formulation | Non-Volatile Contents (%) | Viscosity (mPa·s) | Gel Times (s) |
|---|---|---|---|
| UF | 53.7 | 46.27 | 148 |
| TUF1 | 54.3 | 148.9 | 100 |
| TUF2 | 53.7 | 120.6 | 90 |
| TUF3 | 54.2 | 1019 | 69 |
| TUF4 | — | — | — |
| TUF5 | 54.5 | 162.0 | 124 |
| TUF6 | 54.1 | 203.5 | 89 |
| TUF7 | 54.1 | 459.6 | 83 |
| TUF8 | 52.1 | 72.9 | 154 |
| TUF9 | 56.3 | 5063 | 75 |
| TUF10 | — | — | — |

"—" represented no data for the sample.

On the other hand, the viscosities of TUFs increased very significantly after tannin addition, which can be related to the high original viscosity of tannin/water mixture (Table 4). The extent of increase seems to be dependent on the reaction conditions. For example, the TUF prepared through a longer reaction time or higher reaction temperature will make the viscosity increase appreciably, because these factors lead to higher degree of polycondensation and larger molecular weight.

The general trend for the gel time after modification showed a clear drop by different extents, indicating elevated activity with more tannin incorporation. However, TUF8 exceptionally exhibited a gel time as high as 154 s, which can be linked to low tannin content or low extent of polycondensation.

### 3.2. Investigations Using FT-IR

Figure 2 shows the collected FT-IR spectra to identify the differences between UF and TUF resins, whereas assignments of the relevant peaks are also listed in Table 5. As seen in Figure 2, the broad peak existing in all spectra curves at about $3316\ cm^{-1}$ was related to the -OH stretching vibration. The UF resin shows typical absorption peaks that $1640\ cm^{-1}$ was associated with the C=O stretching of primary amide, $1543\ cm^{-1}$ was related to the C=N stretching of secondary amines, and $1255\ cm^{-1}$ was related to the =C-N or =CH=N stretching of tertiary cyclic amides. Besides, absorption peaks at $1385\ cm^{-1}$ can be attributed to C-H mode in $CH_2$ and $CH_3$ $1364\ cm^{-1}$ was related to C-N stretching of $CH_2$-N; $1136\ cm^{-1}$ was related to C-O-C stretching of aliphatic ether; and $1004\ cm^{-1}$ was related to C-O stretching of methylol groups [2,43]. In case of TUF resins, characteristic

absorption peaks of amide I, amide II, and amide III have shifted from 1644, 1543, and 1255 cm$^{-1}$ to 1652, 1540, and 1261 cm$^{-1}$, respectively, due to the insertion of tannin that changed the chemical environment of the resin. Theoretically, due to the urea reduction, the absorption peaks at 1640 and 1364 cm$^{-1}$ decreased while the absorption peak at 1640 cm$^{-1}$ increased. This may be caused by the superposition of the C=O stretching vibration of the aromatic ring [44]. Moreover, more absorption peaks can be observed that 1458 cm$^{-1}$ was related to C=C stretching in the aromatic ring; 1172 cm$^{-1}$ was related to C=C stretching in the aromatic ring; 899 cm$^{-1}$ was related to C-O stretching, and C-C out of plane bending vibration of the aromatic ring [45,46].

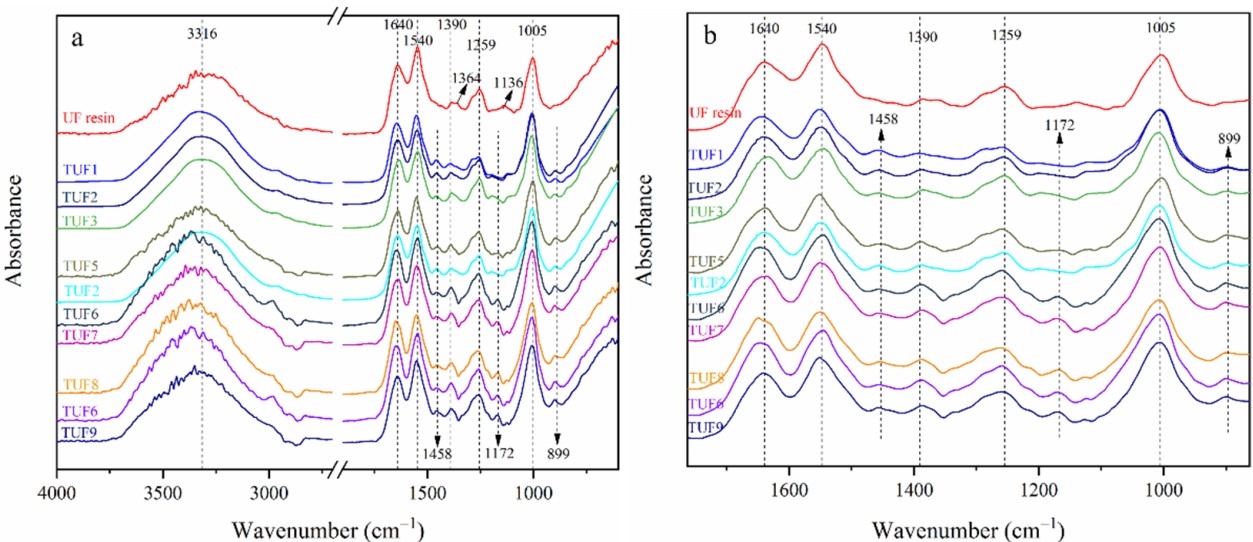

**Figure 2.** FT-IR spectra of UF and TUF resins. (**a**) the full spectra; (**b**) the partial enlarged spectra.

**Table 5.** Assignments of the relevant peaks in FT-IR spectra of UF and TUF resins.

| Absorption Band (cm$^{-1}$) | Chemical Structure Assignment | Assignment | |
| --- | --- | --- | --- |
| | | UF Resin | TUF Resin |
| 3500~3200 | O-H and N-H stretching vibration | 3316 | 3382 |
| 1660~1630 | C=O stretching of primary amide or C=O stretching of aromatic ring | 1644 | 1652 |
| 1560~1550 | C=N stretching of secondary amines | 1543 | 1540 |
| 1600~1400 | C=C stretching in aromatic ring | — | 1458 |
| 1400~1380 | C-H mode in CH$_2$ and CH$_3$ | 1385 | 1390 |
| 1370~1360 | C-N stretching of CH$_2$-N | 1364 | — |
| 1260~1250 | =C-N or =CH=N of tertiarycyclic amides | 1255 | 1261 |
| 1150~1130 | C-O stretching of C-O-C stretching of aliphatic ether or C-O stretching and C-C bending of arene | 1136 | 1172 |
| 1020~1000 | C=O stretching of methylol group | 1004 | 1011 |
| — | aromatic C-H out of plane bending vibration | — | 899 |

"—" represented no data for the sample.

### 3.3. Investigations Using Solid-State $^{13}$C NMR

According to the previous study, the $^{13}$C NMR spectra of UF resin had four regions about carbon peaks [47–49]: 155~170 ppm attributed to the carbonyl carbon of urea and its derivatives; 69~95 ppm attributed to the methylene ether bond; 65~72 ppm attributed to hydroxymethyl; 45~60 ppm attributed to methylene. To confirm the variation of TUF resin, Figure 3 shows the solid-state $^{13}$C NMR of cured TUF resin and the relevant peak assignment was listed in Table 6 to compare the difference between UF resin and TUF resin. It can be observed from $^{13}$C NMR data that the peaks at 158.9, 72.8, and 53.9 ppm refer to carbons of the amide, hydroxymethyl, and methylene, respectively. This structure

indicated that there are still many amide bonds in the cured structure of TUF resin, and the connection mode is still mainly methylene bridges. In addition, the peak at 141 ppm represents the double bond connection between the carbon on the benzene ring and the surrounding oxygen, and the peak at 127 ppm reveals the connection between the carbon on the benzene ring and the methylene carbon resulting from substitution on the benzene ring [44,50]. It is clear that the amide groups will reduced (easily hydrolyzed by water) in the system when tannin participates in the copolymerization with methylol urea, which is also the reason for improvement of water resistance of the TUF resins. Moreover, due to the characteristics of tannin that quickly reacts with formaldehyde, it reduces the free formaldehyde in the resin system.

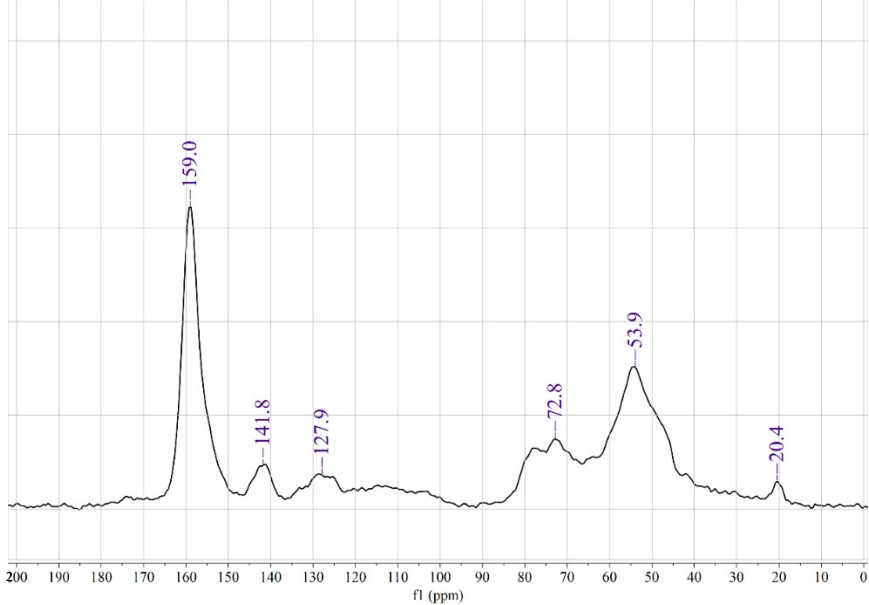

**Figure 3.** Solid-state $^{13}$C NMR spectrum of TUF6 resin.

**Table 6.** Assignments of the relevant peaks in $^{13}$C NMR spectra of UF and TUF resins.

| Chemical Shift (ppm) | | Chemical Structure Assignment |
| --- | --- | --- |
| UF Resin | TUF Resin | |
| 155~170 | 159.0 | Carbonyl carbon of urea and its derivatives |
| — | 141.8 | Double bond connection between the carbon on the benzene ring and the surrounding oxygen |
| — | 127.9 | Connection between the carbon on the benzene ring and the methylene carbon |
| 69~95 | 72.8 | Methylene ether bond |
| 65~72 | — | The carbon of hydroxymethyl |
| 45~60 | 53.9 | The carbon of methylene |
| — | 20.4 | Rotating sideband peaks |

"—" represented no data for the sample.

### 3.4. DSC Investigations

The DSC analysis is often performed to check the curing reaction of thermosetting resins. The differential calorimetric scan of UF and TUF resins are displayed in Figure 4a–c and the peak value representing curing temperature were listed in Figure 4a–c. As mentioned earlier, the reaction temperature and time will affect the degree of co-polycondensation. The peak curing temperature of the resin first decreased and then increased with the deepening of the reaction. When the degree of reaction is low, the required curing temperature is higher, but when the degree of reaction gradually deepens, the curing peak temperature also increases. The reason for that is presumably the molecular cross-linking points of the curing system increased and the molecular chain movement

became more complicated, which caused a higher curing temperature. The changes in peak curing temperature on a single factor of the reaction temperature and reaction time confirmed this assumption. In addition, by reducing the amount of incorporated tannin, the curing temperature of the resin does not change much. However, increasing the amount of tannin causes the opposite, which is due to the high curing temperature of tannin itself.

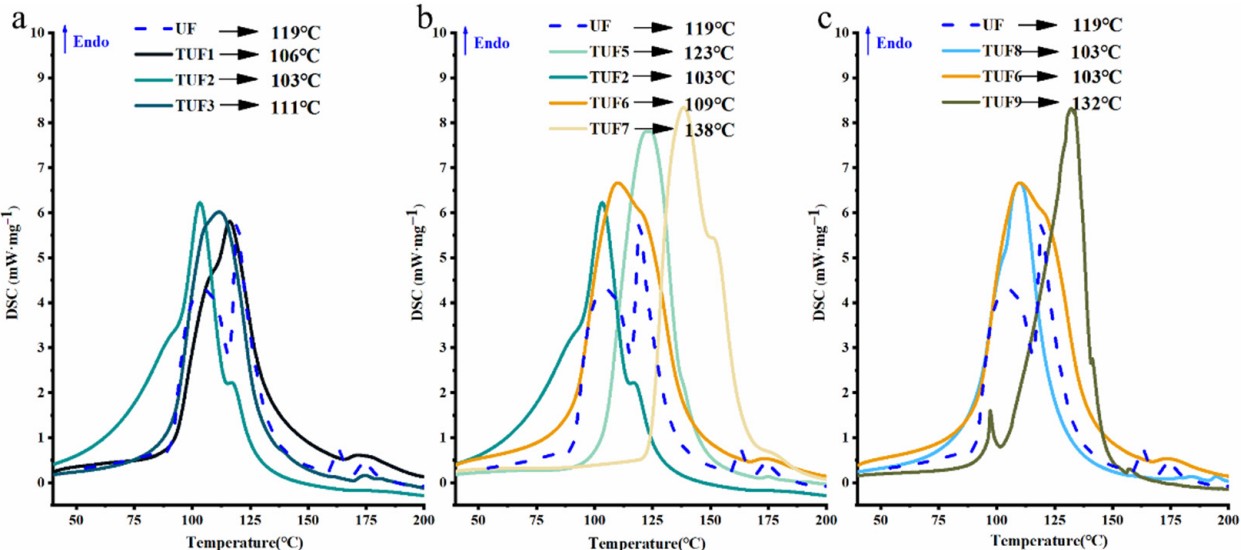

**Figure 4.** DSC of UF and TUF resins prepared under varied reaction conditions. (**a**) reaction temperature; (**b**) reaction time; (**c**) tannin addition.

### 3.5. TG Investigations

The thermal stability of resins was evaluated by TGA to assess their susceptibility to degradation, and the results are given in Figure 5a–c. As seen the TG and DTG curves in Figure 5, the thermal degradation commonly occurs in three stages at 30~105 °C, 105~400 °C, and 400~600 °C. The weight loss in the first step of degradation occurs by eliminating any absorbed water. The second step is the main stage of weight loss that the resins encounter. From the DTG curves, there are several steps in case of UF resin: the peak at 168 °C refers to the decomposition of terminal methylols of the resin into formaldehyde and water. The peak at 264 °C was attributed to the cleavage of the methylene ether bond with the release of formaldehyde, and the peak at 306 °C was attributed to the decomposition of methylol urea [51,52]. Additionally, during the stage of 105~400 °C, the weight loss of UF resin is up to 77.31%. Compared to UF resin, it is also observed a weightless peak at around 168 °C for TUF resins in DTG curves, but the weight loss rate is lower than UF resin. It can be explained that the introduction of the tannin structure caused a decrease in free methylols. Additionally, the weight loss of modified TUF resins reached the maximum rate in the temperature at an interval of 250~270 °C and their weight loss was only around 60% to 64%, lower than UF resin, indicating modified TUF resins acquired outstanding thermal stability with respect to UF resins in the second thermal stage. In addition, the weight residual of UF resin at 600 °C reached 15.7%, and all the TUF resins were in the range of 23%~29%. These results led us to the conclusion that TUF resins had better thermal stability than UF resin.

### 3.6. Effect of Preparation Temperature of TUF on the Shear Strength of Plywood

The bonding performance of TUF resins is influenced by the reaction temperature as can be seen from Figure 6. It was observed that TUF resins gave slightly lower shear strength as compared with UF resin, under cold-wet conditions, but performed better in warm and boiling water. Interestingly, the wood failure of samples in cold-wet conditions reached a value above 90%, close to plywood bonded by control UF resin. As it is known, when the bonding force of the resin is greater than the strength of the wood itself, the

shear strength does not fully reflect the bonding quality of the resin, but it shows that the plywood has a higher rate of wood failure after shear test [53]. Thus, in order to compare the bonding quality between the UF and TUF resins, shear tests were conducted on plywood after conditioning in warm and boiling water. The results revealed that the shear strength values of TUF bonded plywood were improved greatly by the reaction temperature going up, considering that plywood bonded with UF exhibited complete failure within 3 h in boiling water. Among them, TUF3 tested in warm-water conditions reached a strength of 1.48 MPa, more than 45% higher with respect to UF resin, and boiling water conditions reached 0.61 MPa. These results are ascribed to the high temperature, which promoted the reaction between tannin and formaldehyde, resulting in a more stable structure with respect to low temperature conditions. It can concluded that the reaction between tannin and formaldehyde is more favored under higher temperature [54]. In addition, a high temperature can promote the reaction between tannin and methylol urea as well.

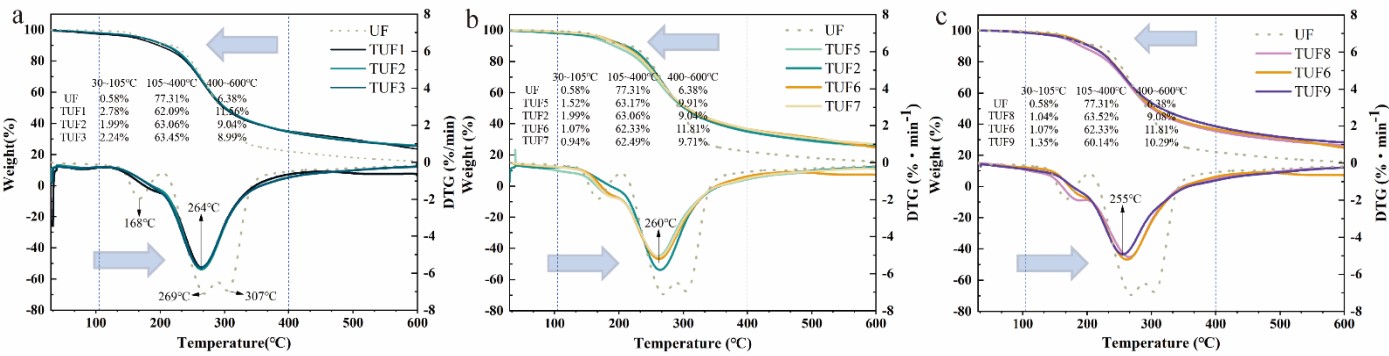

**Figure 5.** TG and DTG thermograms of UF and TUF resins prepared under varied reaction conditions. (**a**) reaction temperature; (**b**) reaction times; (**c**) tannin addition.

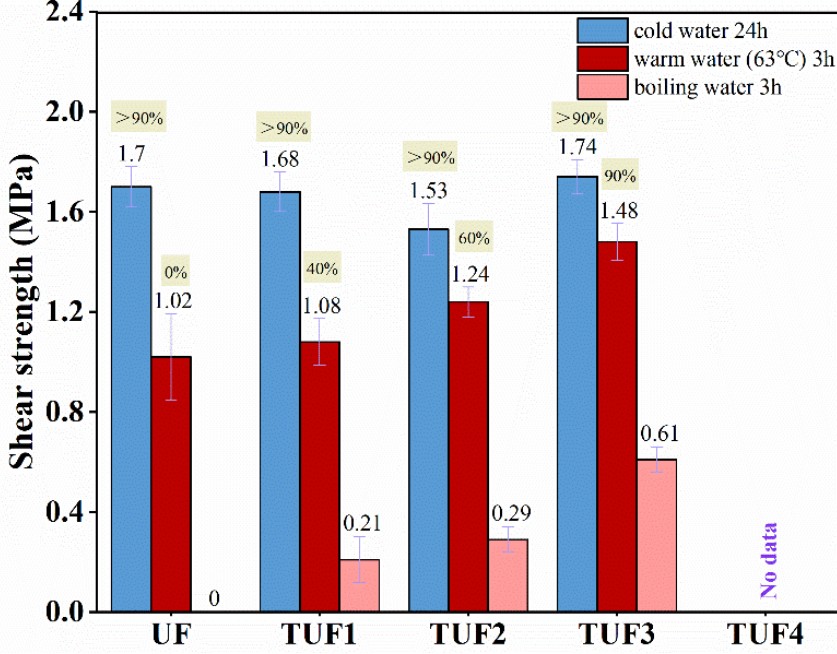

**Figure 6.** Shear strength values of plywood specimens bonded with resins prepared at different reaction temperatures (the highlight indicates the wood failure rate).

## 3.7. Effect of Reaction Time for TUF Preparation on the Shear Strength of Plywood

The bonding strength of plywood bonded using TUF resins is likely to be affected by reaction time of resins preparation. The results of plywood bonded with resins prepared at different times of polycondensation are shown in Figure 7. Apparently, the plywood

bonded with TUF resins showed a trend of decreased shear strength, compared with typical UF resin, after soaking in cold water for 24 h, even for TUF7, which is prepared at the same reaction time as for UF resin (2 h). The shear strength goes down significantly after cold-water treatment, especially for the formulations prepared under a short reaction time, 0.5 h, which may be insufficient for copolymerization between the reaction components to take place. However, the best performance was obtained after 1 h of reaction (TUF2), as can be shown in Figure 7. However, a similar wood failure rate to UF resin in all TUF samples was observed.

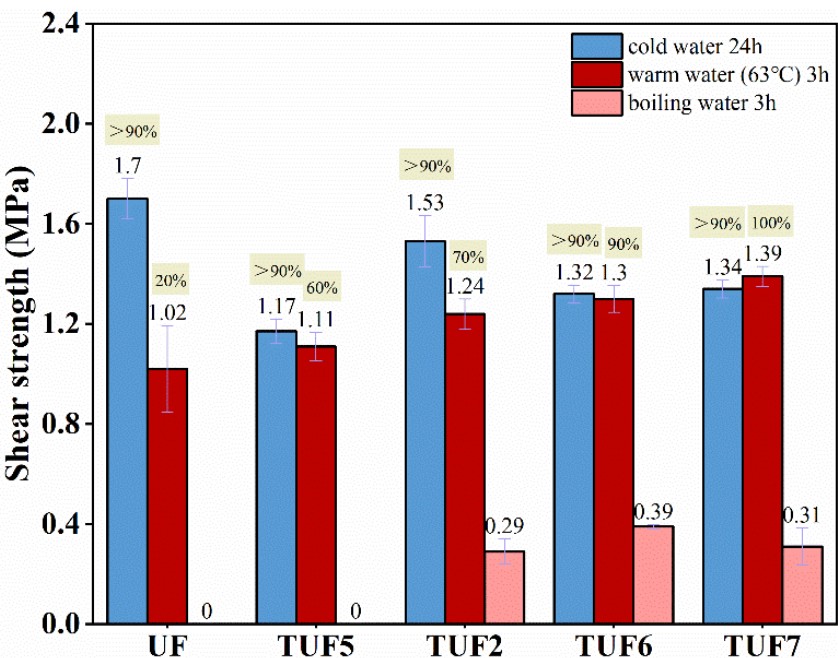

**Figure 7.** Shear strength of plywood specimens bonded with resin prepared via different reaction time (the highlight indicates the wood failure rate).

In case of the warm-water treatment, it can be inferred that the shear strength of plywood specimens showed an increasing trend with the extension of reaction time. When the reaction time was 2 h, the shear strength of TUF resin reached 1.39 MPa, 36% higher than UF. Additionally, most of the TUF resins-based plywood exhibited reasonable shear strength after being soaked in boiling water for 3 h, except TUF5, which signifies that the reason belong to that the copolymerization was not profound.

*3.8. Effect of Tannin Addition on the Shear Strength of Plywood*

Tannin addition can increase the environmentally friendly nature of the modified resins. Most importantly, it can strengthen the bonding performance of TUF resins under either warm water or boiling water condition as compared to UF resin. Figure 8 demonstrates the shear strength of plywood bonded using TUF after cold-water soaking is still lower than that of UF resin, which is similar to the samples obtained with resins prepared via different reaction temperatures and times. However, the bonding performance of TUF-derived plywood under warm or even boiling water soaking conditions was superior to that of UF resin remarkably. This can be attributed to the substitution of second batch of urea using tannin, thus increased the molar ratio of F/U, which has benefits on the formation of dimethylourea, which is the main component to crosslink between resin layer and wood. This causes the performance of TUF resins to improve significantly considering the parallel strengthening effect on this network structure by the tannin. However, an increase in the molar ratio of F/U has a negative effect that often generally leads to an increase in the amount of formaldehyde emission [55]. However, as this was not the case, this indicates that the tannin was incorporated effectively into the network structure by consuming a

large amount of hydrophilic methylol urea, therefore the cured TUF-based formulations acquired better resistance to warm or boiling water than UF resin. Furthermore, the structure of tannin itself increases the rigidity of the resin system and improves the bonding quality [36,44]. Last, it is thought that the decomposition provoked in hot/boiling water to the plywood bonding can be dynamically reversed via a compensation reaction that involves alternative bonds formation.

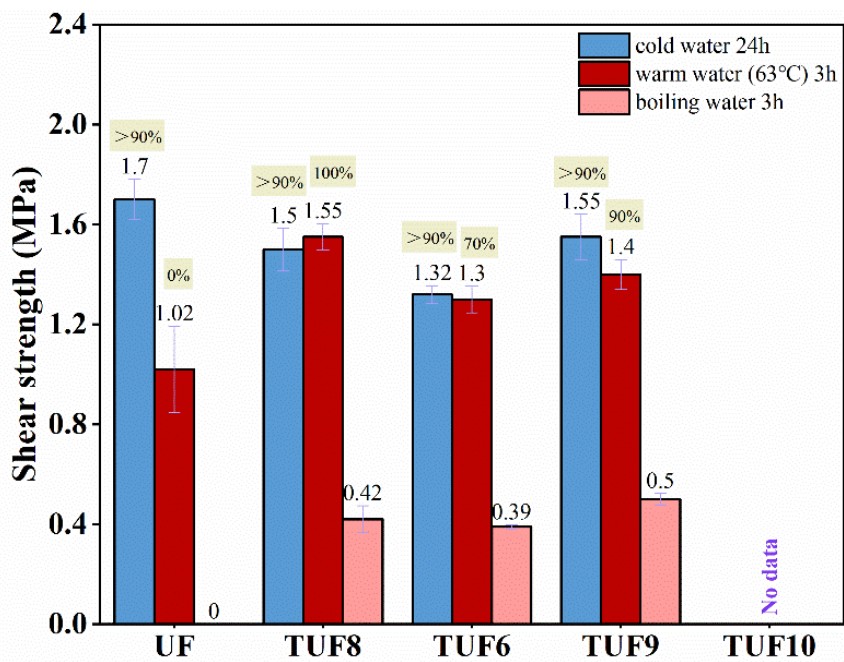

**Figure 8.** Shear strength of plywood specimens bonded with resins prepared with different tannin additions (the highlight indicates the wood failure rate).

### 3.9. Effect of Diferent Plasma Treatment on Veneer Surface for Shear Strength of Plywood

TUF6 resin was selected as a representative resin for bonding, pressure was applied for plywood preparation using veneer treated by different plasma techniques, and the strength was followed up as a function of plasma treatment type (Figure 9a). According to Figure 9a, after the veneer was treated with SAC plasma, the shear strength of plywood specimens improved by 32% and 21%, compared to the untreated samples after soaking in cold-water and warm-water, respectively, while the shear strength was slightly reduced after boiling-water soaking. However, instead, through CLP and JTLP plasma treatments, the shear strength after cold-water test decreased by 8% and 9%, respectively, whereas the shear strength of warm-water test decreased by 9% and 27%, respectively. Under boiling water, the plywood specimens achieved excellent performance, reaching 0.80 MPa and 0.85 MPa, respectively, which translates improvements into 105% and 118%, respectively, with respect to untreated plywood samples. It can be found that the shear strength results after the three plasma treatments showed an unusual pattern, such that the results of cold-water and warm-water presented a decreasing trend, but the result of boiling water increased. The possible reason is that the surface roughness of poplar veneer was improved under the treatment of SAC plasma, which is conducive to the penetration of resin on the surface of the veneer and improved the quality of gluing, but formed little effective surface free radicals, which have no benefit for the resistance to boiling water; however, CLP and JTLP plasma treatment of veneer may lead to excessive etching of the cell wall on the surface of poplar veneer [56], resulting in a decrease in the mechanical strength of the interface between poplar veneer and resin [57], thereby affecting its mechanical strength under cold and warm water tests, but CLP and JTIP plasma treatment can form a large number of free radicals such as hydroxyl and carboxyl groups on the surface of the veneer,

as shown in Figure 9b, which can effectively chemically cross-link with the resin structure and enhance its bonding quality in the boiling water test [58–60].

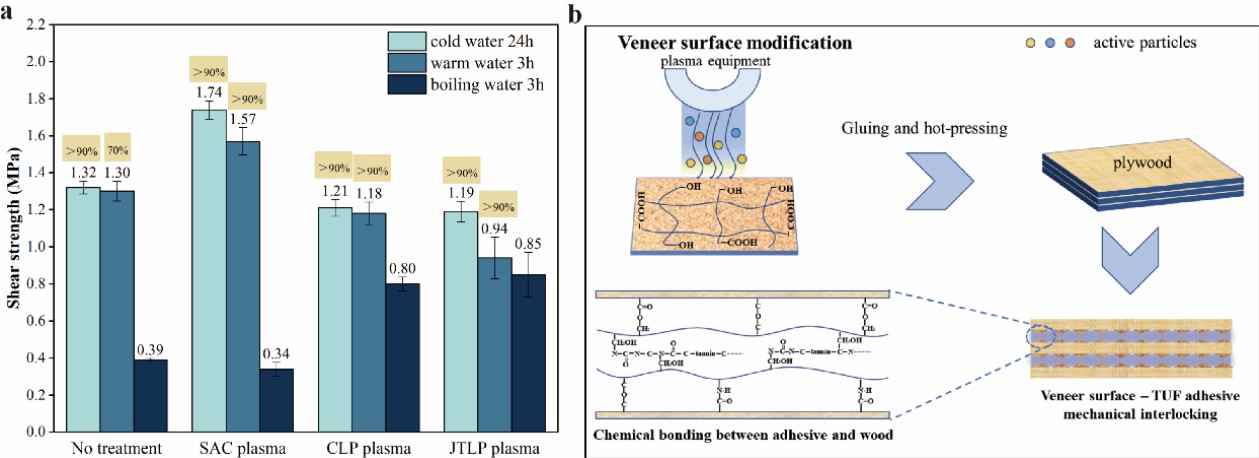

**Figure 9.** (**a**) Shear strength of plywood specimens prepared with veneer exposed to varied regimes of plasma treatments using different equipment; (**b**) Plasma illustration (the highlight indicates the wood failure rate.

### 3.10. Formaldehyde Emission from UF and TUF Resins

The formaldehyde emission of plywood is one of the criteria for judging its quality. Reducing the formaldehyde emission is also an essential concern in the field of wood-based panels [61,62]. Due to the reactivity with formaldehyde, tannin is often used as a formaldehyde scavenger for resins [45,62]. In this study, without the second batch of urea, the resin will have a high mole ratio of F/U, which is most likely to trigger a high free formaldehyde level in the resin system [55]. However, Figure 10 illustrates that the formaldehyde emission from TUF resins was lowered in most plywood samples with respect to UF, which can be directly related to the presence of tannin. It is obvious from the figure that the formaldehyde emission from the plywood specimens increases as the reaction temperature during the resin synthesis increases. However, extending the reaction time of the resin, the formaldehyde emission of the plywood specimens shows a tendency to decrease first and then increase. When the reaction proceeded for 1.5 h, the formaldehyde emission reached a level, as much as 1.42 mg/L, which is 39% lower with respect to UF resin. Furthermore, the increased tannin content also helps to reduce the amount of formaldehyde emission from plywood. When increasing the amount of added tannin to 25 g, the value of formaldehyde emission reached 1.3 mg/L, which is 44% less than that of UF resin. In general, the formaldehyde emission of plywood will not only be affected by the temperature and time of resin synthesis, but also by the amount of added tannin. The higher the added tannin, the lower the formaldehyde emission. However, it should be clear that too much tannin addition can cause gelation.

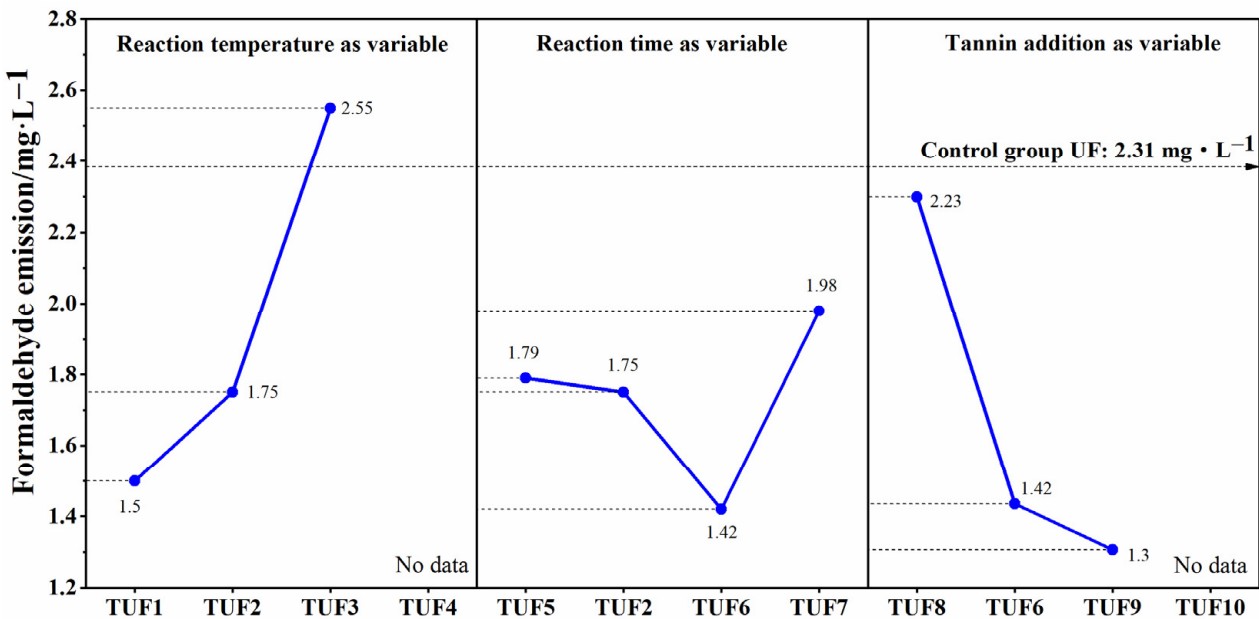

**Figure 10.** Formaldehyde emission level from the different prepared resins.

## 4. Conclusions

The method reported in this study by adding a natural source such as bayberry tannin, can effectively improve the bonding properties of conventional UF resin, especially in terms of water resistance, which is considered a green renewable approach for the synthesis of TUF resins. The joining of tannin introduced some aromatic rings and enhanced the structural rigidity of TUF resins, while improving the stability against hydrolytic degradation. The curing temperature of TUF resins fluctuated around that of UF resin, although TUF resins exhibited higher thermal stability with respect to UF resin.

The preparation method of TUF resin reported in the current work provides a broadened way to use green renewable materials for industrial applications and promotes the modification of UF resin to overcome its drawbacks.

**Author Contributions:** X.Z. and X.C. conceived and designed the experiments and revised the manuscript as well; J.P. performed the experiments; J.Z. tested the partly properties, G.D. and H.E. analyzed the parts of data; J.P. and X.C. wrote the manuscript; X.C., G.D. and X.Z. supervised this work. All authors have read and agreed to the published version of the manuscript.

**Funding:** This research was funded by the National Natural Science Foundation of China (NSFC 31971595), the Key Program of Applied and Basic Research in Yunnan Province (Grant No. 202101AS070008). This work was also supported by the "Ten-thousand Program" youth talent support program and Yunnan Provincial Reserve Talents for Middle & Young Academic and Technical Leaders (2019HB026), and the 111 project (D21027).

**Institutional Review Board Statement:** Not applicable.

**Informed Consent Statement:** Not applicable.

**Data Availability Statement:** Not applicable.

**Acknowledgments:** The authors thank Chunlei Dong and Yunwu Zheng for the guidance of DSC and mechanical test experiments.

**Conflicts of Interest:** The authors declare no conflict of interest.

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
