# Peer review of "Characterization on the Copolymerization Resin between Bayberry (Myrica rubra) Tannin and Pre-Polymers of Conventional Urea–Formaldehyde Resin"

_forests, doi:10.3390/f13040624_

Round 1
Reviewer 1 Report
Comments and Suggestions are bellow
Manuscript ID : forests-1664318
They authors worked on a novel tannin modified UF resin was reported by substituting a part of the urea with tannin, they stydied the impact of the addition of tannin on reaching a state of effective co-polymer ization with urea-formaldehyde prepolymer and avoidance of the classical disadvantages of urea-formaldehyde resins, different plasma surface modification regimes were be adopted to pre-treat the wood veneer, which is expected to enhance the performance of plywood samples bonded with TUF resin. They obtained interesting results. this paper is accepted with following minors revisions.
- The title of the paper is not good, the authors must improve it
- the abstract should be improved by synthesizing it further
- line 34, authors can improve the sentence "...resin will grow at an annual rate of over..."
- line 36, «….. as the binder….. » or « ……as a biender….. » ?
- lines 46-47, we can improve this sentence by removing "impose" and "in"
- the sentence « the toxic formaldehyde » can be changed
- line 85, precize the meaning of "TUF" sigle
- material section (2.1), line 85 to 92, there are some acronyms such as CP and others whose meaning are not specified
- Authors must add references in the Synthesis of UF resin (2.2) section , same for 2.4. section ( Evaluation of the resin propertie), 2.5 section , 2.6 ; 2.7 ;2.8 and 2.9 sections
- line 195, the sentence « …The rapid gelation in case of TUF4 and TUF10.. » must be improved.
- why did the authors recall the paragraph from line 213 to 221?. these results are known. in my opinion their results and discussions of that section beguin from line 222
- the peak located at 3000 and 3500 cm-1 of figure 2 correspond to what. ?
- you can synthetised in a table, the Assignments of the relevant peaks in 13C NMR spectra of UF and TUF resins as you done in table 5.
- the peak located at 20 ppm of figure 3 can be attributed what?
- you can improve the sentence between the lines 263 and 264
- in DSC investigations result section, what did the peak of each figure 4a-C stand for?
- the authors use two word in paper : resin and adhesive. They can choice one
- The legends of figure 5 don’t show exactly the TG thermograms of TUF adhesives. Ameliore it.
- change the title of figure 5
- what represent the DTG curves of Figures 5a-c?. explain they in the text
- give they differents steps in case of TUF adhesive as you done in lines 280 and 281.
- Explain the influence of the masses 15g, 20g and 20g on the behavior of the resins placed as legend on figure 5 c. Same thing for durations 05h-2h on figure 5b and temperatures 60°C, 70°C and 80°C of figure 5a
- line 324, why did the best performance was obtained after 1 h of reaction (TUF2) as can be shown in Figure 3 and not after 2h ?
- Improve the sentence of line 350
- change the colors of figures 6, 7 and 8
- In figure nine, they said no data for TUF10. Why ?
- Improve the sentence of line 361
- are lines 361 to 365 in their place?. this falls under materials and methods section
- lines 369 to 372, you said « …However, instead, through CLP and JTLP plasma treatments, the shear strength after cold-water test decreased by 8% and 9%, respectively, whereas the shear strength of warm-water test decreased by 9% and 27%, respectively » why ? explain.
Author Response
Dear reviewer,
Thank you very much for your positive comments. I already gave the response point-by-point as follows:
1. Comment:The title of the paper is not good, the authors must improve it.
A: Thank you for your suggestion. We have revised the title.
2. Comment: the abstract should be improved by synthesizing it further
A: Thank you for your comments. The abstract part have been improved.
3. line 34, authors can improve the sentence "...resin will grow at an annual rate of over..."
A: Thanks, revised.
4. line 36, «….. as the binder….. » or « ……as a binder….. » ?
A: Thank, revised.
5. lines 46-47, we can improve this sentence by removing "impose" and "in"
A: Thanks for your suggestion. revised.
6. the sentence « the toxic formaldehyde » can be changed
A: Thanks, revised.
7. line 85, precize the meaning of "TUF" sigle
A: Thanks, we have added some essential information of TUF.
8. material section (2.1), line 85 to 92, there are some acronyms such as CP and others whose meaning are not specified
A: We are sorry for our mistake, and we have added some detailed explanation on them.
9. Authors must add references in the Synthesis of UF resin (2.2) section, same for 2.4. section ( Evaluation of the resin properties), 2.5 section , 2.6 ; 2.7 ;2.8 and 2.9 sections
A: Thank you for your comments, the references have added in section of 2.2, 2.4, and 2.9. We have given the measurement details in section of 2.5, 2.6, 2.7 and 2.8, thus there is not necessary to cite references again.
10. line 195, the sentence « …The rapid gelation in case of TUF4 and TUF10.. » must be improved.
A: Thanks, revised.
11. why did the authors recall the paragraph from line 213 to 221?. these results are known. in my opinion their results and discussions of that section beguin from line 222
A: Thanks for your suggestion. We have removed the sentence between 213 and 221 in original version.
12. the peak located at 3000 and 3500 cm-1 of figure 2 correspond to what. ?
A: We are very sorry for our negligence, and we have added the interpretation in 3.2 section, the broad peak existing in all spectra curves at about 3316 cm-1 is related to the -OH stretching vibration.
13. you can synthetised in a table, the Assignments of the relevant peaks in 13C NMR spectra of UF and TUF resins as you done in table 5.
A: Thank you for your suggestion. We have added the detailed information as the form of Table 5 in this part.
14. the peak located at 20 ppm of figure 3 can be attributed what?
A: Thank you for your comment. We have explained in table 6, 20 ppm was interpreted to Rotating sideband peaks.
15. you can improve the sentence between the lines 263 and 264
A: Thanks, revised.
16. in DSC investigations result section, what did the peak of each figure 4a-C stand for?
A: We are very sorry for our negligence, we have added description in the 3.4 section.
17. the authors use two words in paper: resin and adhesive. They can choice one
A: Thank you for your suggestion. We have revised.
18. The legends of figure 5 don’t show exactly the TG thermograms of TUF adhesives. Ameliore it.
A: Thank you for your suggestion. We have ameliorated it.
19. change the title of figure 5
A: Thank you for your suggestion. We have changed the title of figure 5.
20. what represent the DTG curves of Figures 5a-c?. explain they in the text
A: Apologies for this negligence. We have made the change in figure and the text which make the reader easier to understand Figures 5a-c.
21. give they differents steps in case of TUF adhesive as you done in lines 280 and 281.
A: Thank you for your suggestion. We have carefully reviewed the description in 3.5 section and made a further distinction between the TGA curves of UF resin and TUF resins
22. Explain the influence of the masses 15g, 20g and 20g on the behavior of the resins placed as legend on figure 5 c. Same thing for durations 05h-2h on figure 5b and temperatures 60°C, 70°C and 80°C of figure 5a
A: Thank you for your suggestion. We have reworked the explanation for the figures 5a-c, and focused on the comparison between thermal behavior of UF resin and TUF resins. However, the TUF resin samples had similar thermal stability that we did little explanation about the influence of reaction conditions.
23. line 324, why did the best performance was obtained after 1 h of reaction (TUF2) as can be shown in Figure 3 and not after 2h ?
A: Apologies for our careless. The original Figure 3 should be Figure 7, and in this paragraph, we compared the performance in cold water 24h which the TUF2 have best performance as shown in Figure 7.
24. Improve the sentence of line 350
A: Thank you for your suggestion. We have revised.
25. change the colors of figures 6, 7 and 8
A: Thank you for your suggestion. We have made the change.
26. In figure nine, they said no data for TUF10. Why ?
A: Happy to answer your question. As we explained in the 3.1 section, TUF4 and TUF10 would occur the gelation accident during the synthesis, but for the purpose to keep the continuity of data, we kept the legend of TUF10.
27. Improve the sentence of line 361
A: Thanks, revised.
28. are lines 361 to 365 in their place?. this falls under materials and methods section
A: We thank the reviewer for pointing this out. We have revised.·
29. lines 369 to 372, you said « …However, instead, through CLP and JTLP plasma treatments, the shear strength after cold-water test decreased by 8% and 9%, respectively, whereas the shear strength of warm-water test decreased by 9% and 27%, respectively » why? explain.
A: Thank you for your comments. We have add the new explanation in the 3.9 section and introduced two citation ([57] and [58]) to prove it.
Reviewer 2 Report
generally, this paper was written with good style and more data. the overall of this paper is good and it is suitable to be accepted for publication in this journal after making some major and minor revisions....

Author Response
Dear reviewer,
Thank you for your valuable comments, I gave the response point-by-point as follows:
1. In the title, the word (bioresourced) and (partially substituting urea) may be removed. And this title contains some disconnections in the meaning such as in (the copolymerization process of biosourced bayberry tannin) i.e. the title did not show that the copolymerization of bayberry tannin was copolymerized with any comonomer….The sentence (for enhancing the characteristics and performance) did not point out the enhancement in characteristics and performance of any material or product??? The title is very long and must be rewritten.
A: Thank you for your excellent suggestion, we have modified the title of manuscript.
2. The names of the last two authors did not contain superscript numbers, such as that found in the first three authors. Hisham Essawy 3 or Hisham Essawy 2.
A: We are sorry for this mistake; we have corrected it in our manuscript.
3. In the abstract, the authors in most articles generally write the main synthesis process and characterization methods before the main findings and results. But in this paper, the characterization methods such as FTIR, 13C-NMR, and DSC were found in the last part of the abstract. Therefore, I suggest that the abstract at this state needs to some rearrangement.
A: Thank you for your valuable suggestion. We have improved the abstract part.
4. In the materials section: Bayberry tannin was purchased from Guangxi Wuming Quebracho plant. Some tannin properties such as tannin type and molecular weight as the manufacturer or supplier reported will be more informatics to the interest readers if it added the material section.
A: We gratefully appreciate for your valuable suggestion. We added information about bayberry tannins, such as main structure and tannin purity.
5. In the urea-formaldehyde resin synthesis procedure i.e. (2.2. Synthesis of UF resin), the researchers did not show if there are a purification and drying steps to get pure UF resin and also in TUF resins synthesis. And the practical conditions that are used to removing of unreacted formaldehyde or formic acid salts must be briefly added to this paragraph.
A: Thank you for your question. Since UF resin and TUF resin prepared in this experiment were not dried or purified and used directly in the preparation of plywood, there is the reason why no relevant explanation in this paper.
6. On the page 3 of 16, I suggest this title 2.3. Preparation of tannin modified UF resin (TUF) instead of the long title (2.3.Partial substitution of urea using bayberry tannin for preparation of modified UF resinS(TUF)).
A: Thank you for your valuable suggestion. We had modified it according to your opinion.
7. In Table 1, sample TUF2 is duplicated in table 2. And in Table 2, sample 6 is duplicated in Table 3. Why?
A: We are happy to answer your question. As shown in Table 1, Table 2 and Table 3, this experiment was carried out by means of multi-factor experiments. The experiments listed in Table 1, Table 2 and Table 3 were carried out in turn. The experiment in Table 2 is based on the optimal formula with the comprehensive results of the experiment in Table 1. And Table 3 is the same for this reason. Thus, in order to maintain the continuity of the table factor levels, TUF2 and TUF3 are repeated in Table 2 and Table 3 in turn. This duplication also occurs in the Results and Discussion section.
8. In Table 4, the authors did not discuss the reasons to get relatively very low viscosity for sample TUF8 in spite of it containing a high tannin loading level. And what about the basic properties, thermal properties, formaldehyde emission, shear strength, and performance properties of samples TUF4 and TUF10???
A: Thanks for your good comments. We also prepared the sample TUF4 and TUF 8, but unfortunately, we did not get the results of them due to the gelation accident did occur during the preparation process. So, we could not do some further tests on them. Relevant instructions have been highlighted in 3.1 (Line 215-218)
9. In the 2.4. Evaluation of the resin properties: I think that the authors used a reaction mixture of different materials that were used in the synthesis procedure such as TUF resin, NaOH, unreacted formalin, and water. So the sentence: the non-volatile content of the resins was measured by comparison of resin weight before and after drying…. Did it refer to purified or non-purified resins? According to the sentence (about 1 g of resin was put into the oven at 120±1°C for 2h, then…), the researchers dried the resin at 120oC for 2h while the DSC tests show that the curing reactions in those resins occur after 100oC and in the range of 103-138oC. So I think that the 120oC is too high for drying but it is suitable for curing.
A: Thank you for your question. Like the answer of comment 5, the UF resin and TUF resin was used directly after synthesis without purity. The reason why the non-volatile content test was carried out in 120℃ is that making sure the solvent in the resin is completely evaporated. And this test method and process we used in here is according to the related standard requirement, so, we chose 120℃ as the experiment temperature. And the DSC result show the temperature range located in 103-138℃, explain that during these temperature range, the resin do occur curing behavior, so, 120℃ is an acceptable temperature.
10. Please, let the reader know…did the FTIR and DSC samples contain water or only pure and dry liquid resins?. The researchers must be sure about the DSC results if the curing reaction of UF and TUF is exothermic or endothermic based on these results the (small arrow with Endo or Exo ) attached in the figures of DSC scans.
A: Thank you for your suggestion. The preparation methods of resin samples used for FTIR and DSC samples have added in 2.5 and 2.7 section.
11. The method in the 2.9 Evaluation of formaldehyde emission is not well clear and needs more explanation about formaldehyde emission evaluation, especially this step (Then, they were put right above a glass dish filled with 300 mL of distilled water to absorb…..)
A: Thank you for your suggestion. We have added some detailed information in 2.11 of our revised manuscript.
12. In the measurements section, the subtitle such as 2.5 must be splitting
for each technique; one for FTIR and another for 13C-NMR and shorten
without (Investigations using….). Also the same matter for 2.6.
A: Thank you for your suggestion. We agree and have updated.
13. The first paragraph in the (3.2 Investigations using FT-IR and solidstate 13C NMR) which starts with (Condensed tannin has similar reactivity to phenol when it is polymerized with formaldehyde [25,34,35]. However, due to…) and ended with (….agent between the tannin units.) do not appear as a result, but it is suitable more to introduction part.
A: We gratefully appreciate for your valuable suggestion. We have made correction according to the Reviewer’s comments.
14. 3.3 DSC investigations
The author needs more explanations and discussion about the curing behavior of TUF resin and more focusing on the effect of loaded tannin on the obtained thermal data for DSC. Please, insert a Table with obtained DSC results such as curing (primary, maximum, and final) temperatures, curing enthalpies, and curing rate.
A: We agree with the reviewer that further elaborating on this point using new data would be helpful. However, our experiments are only intended to illustrate the curing temperature of different resins. As a result, full experiments were not carried out and the data suggested by the reviewers, such as curing enthalpy, etc., could not be obtained. For this reason, we chose not to make this change.
15. 3.4 TG investigations
Primary, Figure 5c shows that the TUF resins have lower thermal stability than that pure UF resin. While in the conclusion part, the authors state that the TUF resins exhibited higher thermal stability per that of UF adhesive. Generally, without a table with thermal data extracted from TGA thermograms or curves, as a reader or a reviewer, I cannot see or understand any effect of the tannin in the thermal degradation behavior of UF or TUF. The thermal degradation curves of TUF resins must be discussed with more attention to the effect of the tannin, the matter which does not take well by the authors in this paragraph.
A: We apologize for our negligence. We have added the thermal data in the Fig.5 and discussed the results about UF resin and TUF resin.
16. Please, let me ask the authors, about the samples named TUF4 and TUF10: Why these samples are found in whole the manuscript and in all Figures and tables but without any results?? They must be removing from the samples lists and all figures because they did not tested or evaluated.
A: We apologize that TUF4 and TUF10 were listed puzzling the reader. On the high reaction temperature or high tannin addition condition, the TUF4 and TUF10 occurred the gelation accident during the preparation, resulting to difficult identify the resins characteristics, that is why TUF4 and TUF10 had no data in the article. Relevant instructions have been highlighted in 3.1 (Line 215-218).
17. In references list, the reference number 1, it is without a website title
or journal name.
A: Thanks, revised.
Reviewer 3 Report
The manuscript deals with the investigation and evaluation of the effect of partial substitution of urea by bio-sourced tannin in urea-formaldehyde (UF) resin used for bonding three-layer plywood. In general, the manuscript is well-written, structured and informative, but still needs some minor improvements. Please, see below my comments on your work:
In general, the title (lines 2-4), the abstract (lines 10 to 27) and the keywords (lines 28-29) correspond to the title, aims and objectives of the manuscript. The abstract is well-written and informative, and contains the main findings of the article.
Lines 2-4: The title of the manuscript is relevant, but it seems a bit long to me. Please consider removing the last part, i.e. “for enhancing the characteristics and performance”, since it does not bring any valuable information. Please consider also adding the botanical name of bayberry.
Line 11: “Urea” should not be capitalised, please revise.
Line 18: “and” is not necessary, please delete.
Line 32: “related market reseach…” – related to what? Please revise the statement.
Line 37: “wooden merchandise” is not an appropriate term, please revise, e.g. “wood-based composites”
Line 51: “forest residue sources” is not the most appropriate term, please consider revising it, e.g. by lignocellulosic resources or biomass resources.
Lines 53-54: Please revise the sentence “As a natural polyphenol component collected from plants, tannin which comes next to lignin based on its reserve, can be collected from plants” to avoid unnecessary repetition and provide better meaning.
Lines 55-56: In addition to replacement of phenol in phenol-formaldehyde resins, another viable application of tannin, which has gained a strong attention recently, is the development of non-isocyanate polyurethane resins for wood adhesives, based on lignin or tannin as a substitution of toxic isocyanates. Please check the following relevant references:
https://doi.org/10.1016/j.jece.2021.107053
https://doi.org/10.3390/f12111516
https://doi.org/10.1080/10643389.2018.1537741
https://doi.org/10.1016/j.tca.2018.04.013
Line 59-60: “wood artificial products” does not sound properly, please revise.
Line 82: please replace “urea-formaldehyde” with the common abbreviation “UF”.
Line 83: please revise “will be investigated”, you have already performed the study, please use past simple form, i.e. “was investigated”
Lines 83-84: “will be adopted” – please see the previous comment and revise.
Overall, the Introduction part is well written and informative, and provides relevant information and references on the research topic. However, I’d recommend to extend it based on my recommendations.
Line 90: please explain why did you select exactly bayberry tannin. Is there any particular reason for this? Please explain.
Lines 125-126: please add the respective standard GB/T 14074-2017 also in the references of your manuscript.
General comment to all the equipment used – please provide detailed information about the equipment used, i.e. company producer, city, country.
Line 160: section 2.7. Different plasma treatments for activating the surface of veneers – please explain how did you select the regime parameters (power, treatment time).
Line 175: please add relevant information on the hot press used to fabricated plywood panels. In addition, please justify the selected manufacturing parameters for all types of plywood produced. Why did you increase the press temperature to 180 °C for TUF-bonded plywood?
Line 180: please add the standard GB/T 17657-2013 to the references of the manuscript. Please describe the testing equipment, used for evaluating the shear strength of plywood.
Overall, the Materials and Methods section is well written and detailed, but can be further elaborated based on the comments above.
Line 192: please rename section 3 to Results and Discussion
In general, the results of the study are detailed, informative and properly discussed with relevant research works in the field.
The Conclusion part (lines 411-422) reflects the main findings of the manuscript. In addition to the practical application of your results, I’d recommend to add also the potential for future studies in the field.
The References cited are appropriate to the topic of the manuscript. Inclusion of additional references, especially in the Introduction and Results and Discussion section, will significantly increase the scientific merit of the presented manuscript.
Best regards!
Author Response
Dear reviewer,
Thank you for your comments, I gave the following response point-by-point , please check it.
1. Lines 2-4: The title of the manuscript is relevant, but it seems a bit long to me. Please consider removing the last part, i.e. “for enhancing the characteristics and performance”, since it does not bring any valuable information. Please consider also adding the botanical name of bayberry.
A: Thank you for your suggestion, we have corrected the title.
2. Line 11: “Urea” should not be capitalised, please revise.
A: Thanks, revised.
3. Line 18: “and” is not necessary, please delete.
A: Thanks, revised.
4. Line 32: “related market reseach…” – related to what? Please revise the statement.
A: Thank you for your suggestion. The ambiguous word has been replaced.
5. Line 37: “wooden merchandise” is not an appropriate term, please revise, e.g. “wood-based composites”
A: Thank you for your suggestion. We have corrected some information.
6. Line 51: “forest residue sources” is not the most appropriate term, please consider revising it, e.g. by lignocellulosic resources or biomass resources.
A: Thank you for your valuable suggestion. We have changed the expression.
7. Lines 53-54: Please revise the sentence “As a natural polyphenol component collected from plants, tannin which comes next to lignin based on its reserve, can be collected from plants” to avoid unnecessary repetition and provide better meaning.
A: Thanks for your good comments, we have improved this sentence in our manuscript.
8. Lines 55-56: In addition to replacement of phenol in phenol-formaldehyde resins, another viable application of tannin, which has gained a strong attention recently, is the development of non-isocyanate polyurethane resins for wood adhesives, based on lignin or tannin as a substitution of toxic isocyanates. Please check the following relevant references:
A: Thank you for your introduction to these wonderful research work. According to your suggestion, we have properly cited these articles.
9. Line 59-60: “wood artificial products” does not sound properly, please revise.
A: Thank you for your suggestion. We have revised some expression.
10. Line 82: please replace “urea-formaldehyde” with the common abbreviation “UF”.
A: Thanks, revised.
11. Line 83: please revise “will be investigated”, you have already performed the study, please use past simple form, i.e. “was investigated”
A: Thanks, revised.
12. Lines 83-84: “will be adopted” – please see the previous comment and revise.
A: Thank you for your suggestion. We have corrected the related expression.
Second part
13. Line 90: please explain why did you select exactly bayberry tannin. Is there any particular reason for this? Please explain.
A: There are two main reasons we chose the bayberry tannin in our research. First, bayberry trees are planted in large quantities in China, and bayberry tannin has the advantages of abundant yield and easy availability which if it could apply in practical industry, it will produce huge industrial value; Second, bayberry tannin is a type of condensed tannin with high structural reactivity. And we have more application experience on this kind of tannin.
14. Lines 125-126: please add the respective standard GB/T 14074-2017 also in the references of your manuscript.
A: Thanks, revised.
15. General comment to all the equipment used – please provide detailed information about the equipment used, i.e. company producer, city, country.
A: Thanks for you suggestion. We have added the relevant information.
16. Line 160: section 2.7. Different plasma treatments for activating the surface of veneers – please explain how did you select the regime parameters (power, treatment time).
A: Thank you for your good suggestion. The detailed information of the plasma we used in our research which based on some reports of literatures, such as [39], [40] and [41]. Therefore, the regime parameters was confirmed according to the previous research results as shown in these references.
Round 2
Reviewer 2 Report
OK GOOD WORK